# T2*-Weighted Imaging Performance in the Detection of Deep Endometriosis among Readers with Different Experience: Comparison with Conventional MRI Sequences

**DOI:** 10.3390/diagnostics12071545

**Published:** 2022-06-24

**Authors:** Paolo Niccolò Franco, Simona Annibali, Sara Viganò, Caterina Cazzella, Chiara Marra, Antonella Smedile, Pietro Andrea Bonaffini, Paolo Marra, María Milagros Otero García, Caroline Reinhold, Sandro Sironi

**Affiliations:** 1Department of Radiology, Papa Giovanni XXIII Hospital, Piazza OMS 1, 24127 Bergamo, Italy; sannibali@asst-pg23.it (S.A.); s.vigano36@campus.unimib.it (S.V.); c.cazzella@campus.unimib.it (C.C.); asmedile@asst-pg23.it (A.S.); pa.bonaffini@gmail.com (P.A.B.); marrapaolo87@gmail.com (P.M.); sandro.sironi@unimib.it (S.S.); 2School of Medicine, University of Milano-Bicocca, Piazza dell’Ateneo Nuovo 1, 20126 Milano, Italy; 3Casa Medica Polyclinic Health Center, Via Camozzi 77, 24121 Bergamo, Italy; chiaramarra@casamedica.it; 4Hospital Universitario de Vigo (CHUVI), Estrada Clara Campoamor 341, 36312 Vigo, Spain; milagros.otero.garcia@sergas.es; 5Department of Medical Imaging, McGill University Health Centre, 650 Cedar Avenue, Rm C5 118, Montréal, QC H3G 1A4, Canada; caroline.reinhold@mcgill.ca

**Keywords:** MRI, endometriosis, deep infiltrative endometriosis, MRI protocols, T2*-weighted sequences, blood degradations products

## Abstract

Magnetic resonance imaging (MRI) is an effective technique for the diagnosis and preoperative staging of deep infiltrative endometriosis (DIE). The usefulness of MRI sequences susceptible to chronic blood degradation products, such as T2*-weighted imaging, remains uncertain. The present study aims to evaluate the diagnostic performance of these sequences in addition to the conventional protocol for DIE assessment. Forty-four MRI examinations performed for clinical and/or ultrasound DIE suspicion were evaluated by three readers with variable experience in female imaging. The inter-observer agreement between the reader who analysed only the conventional protocol and the one who also considered T2*-weighted sequences was excellent. The less experienced reader diagnosed a significantly higher number of endometriosis foci on the T2*-weighted sequences compared with the most experienced observer. T2*-weighted sequences do not seem to provide significant added value in the evaluation of DIE, especially in less experienced readers. Furthermore, artifacts caused by undesirable sources of magnetic inhomogeneity may lead to overdiagnosis.

## 1. Introduction

Endometriosis is a chronic gynaecological disorder characterized by the presence of ectopic functional endometrial tissue outside the uterine cavity, which responds to ovarian hormones with cyclic haemorrhages, similar to the normal endometrium [1,2]. This condition affects approximately 10% of women of reproductive age and up to 30–50% of women suffering from symptoms such as dysmenorrhea, dyspareunia, chronic pelvic pain, urinary tract symptoms, and infertility [2,3]. Endometriosis typically manifests in three ways: disease limited to ovaries (endometriomas), superficial peritoneal endometriosis, and deep infiltrative endometriosis (DIE), which is defined as subperitoneal invasion by endometriotic lesions that exceed 5 mm in depth [3,4]. The most common sites affected by DIE are the uterosacral ligaments, torus uterinus, posterior vaginal fornix, rectovaginal septum, rectum, and the urinary tract [4,5,6].

Laparoscopy has been conventionally considered the gold standard for diagnosing endometriosis. However, recent guidelines redefined this indication, underlying that imaging methods have achieved high diagnostic accuracy and recommending invasive diagnostic procedures only in patients with negative imaging findings and/or where empirical treatment was unsuccessful [7].

Transvaginal ultrasonography (TVUS) is usually the first-line imaging technique to assess pelvic endometriosis. Nevertheless, the accuracy of TVUS is limited by the restricted field of view and dependence on operator experience [8,9]. Magnetic resonance imaging (MRI) is recognized as the most accurate noninvasive tool for assessing deep endometriosis and defining pre-surgical planning [3,4,10]. T1-weighted (T1W) fat-suppressed MRI images allow DIE assessment when it manifests as lesions containing hyperintense haemorrhagic/proteinaceous foci. At the same time, non-haemorrhagic fibrotic implants are effectively detected on T2-weighted (T2W) images as hypointense irregular plaques [4,6]. However, MRI diagnosis can be challenging because T1W images are sensitive only to subacute blood degradation products, such as methemoglobin, while chronic blood products may not show typical T1W hyperintensity. Conversely, MRI T2*-weighted (T2*W) sequences are susceptible to chronic blood degradation products such as haemosiderin which are detected as signal void artefacts. For this reason, these sequences are widely used in neuroimaging studies to assess cerebrovascular diseases, haemorrhagic infiltrating lesions, neurodegenerative diseases, cerebral amyloid angiopathy, and vascular malformations. [11,12].

A number of publications have recently suggested that T2*W and susceptibility-weighted MRI can provide added value to detecting ectopic endometrium by identifying haemosiderin deposited during cyclic bleeding [13,14]. Nevertheless, these sequences are currently not included in conventional protocols suggested by international guidelines and are still considered under evaluation or optional [10,15].

The present study aims to assess the diagnostic performance of T2*W imaging in DIE detection among readers with different pelvic imaging expertise and evaluate if these sequences can provide valuable additional information to conventionally used MRI protocols.

## 2. Materials and Methods

The study was conducted in accordance with the Declaration of Helsinki. Clinical and radiological data were anonymized during data collection. The local ethics committee’s review of the protocol deemed that formal approval was not required owing to the observational, retrospective, and anonymous nature of this study.

### 2.1. Study Population

This study included consecutive women who underwent a pelvic MRI for clinical and/or ultrasound (US) suspicion of DIE between December 2020 and August 2021. The inclusion criteria were as follows: (a) women of reproductive age, (b) a previous TVUS examination suggestive of DIE, and/or (c) patients complaining of symptoms compatible with DIE such as dysmenorrhea, dyspareunia, and chronic pelvic pain, especially with a cyclical pattern. Major exclusion criteria were: (a) patients younger than 18 years old, (b) patients with absolute contraindications to MRI (i.e., pacemakers/defibrillator carriers), and (c) poor image quality on the T2*W and/or in the conventional protocol sequences. Data regarding patients’ clinical history (i.e., previous surgery, ongoing therapies, menstrual cycle phase) were collected through a screening questionnaire before the MRI examinations.

### 2.2. MRI Protocol

Pelvic MRI examinations were performed on a 3T scanner (Discovery MR750w GEM, GE Healthcare, Little Chalfont, UK), with the patient lying supine and using an anterior 16-channel pelvic-array coil (GE Healthcare, Waukesha, WI, USA). Patients were asked to fast for six hours and empty their bladder one hour before the MRI. An anti-peristaltic agent (Hyoscine butylbromide 20 mg/mL, Boehringer Ingelheim, Ingelheim am Rhein, Germany) was administered by intramuscular injection 15 min prior to the examination. Ultrasound gel (Aquasonic 100, Parker Laboratories, Fairfield, NJ, USA) was instilled into the rectum or the vagina (200 and 50 mL, respectively) in patients with prior US and/or clinical suspicion of involvement of these structures. No rectal preparation was required. MRI examinations were performed regardless of the patient’s menstrual cycle phase. The conventional MRI protocol included: (*a*) an axial single-shot fast spin-echo (SSFSE) 2D T2-weighted sequence, with a 5 mm slice thickness and a large field of view (FOV) extending from the renal hila to the pubic bone; (*b*) multiplanar fast relaxation fast spin-echo (FRFSE) 2D T2-weighted sequences (orientated in relation to the uterine long axis on axial, sagittal and coronal planes) with a 3 mm slice thickness and a small FOV dedicated to the area of interest; (*c*) 3D fast spoiled gradient recalled echo (FSPGR) T1-weighted sequences with and without fat saturation obtained using the Dixon technique on axial plane, with a 3 mm slice thickness and a FOV covering the whole pelvic area. The overall duration of the conventional protocol was 20 min. T2*W images were obtained with multi-echo recombined gradient echo (MERGE) technique in the axial plane, with a 4 mm slice thickness and a large FOV, covering the whole pelvis, with a total imaging time of 5 min. No dynamic post-contrast sequences were acquired, according to international guidelines [10,15]. The detailed MRI protocol used in this study is provided in Table 1.

### 2.3. Image Analysis

The MRI examinations were independently evaluated by three radiologists with varying years of experience in gynaecological imaging. *Reader 1*, a radiologist highly experienced in gynaecologic imaging (13 years), analysed the conventional MRI sequences with the additional contribution of T2*W sequences. *Reader 2*, with *3* years of experience in gynaecologic imaging, assessed the presence of endometriotic lesions only on conventional sequences and blinded to T2*W images. *Reader 3*, who had 1 year of general experience in MRI, evaluated both the conventional and T2*W sequences after a 3-week training period in endometriosis imaging. The three readers assessed the presence or absence of endometriosis using a checklist of pelvic structures (*n* = 22) typically involved in endometriosis (Table 2) and reported the MRI signal for each detected lesion.

Endometriotic lesions were defined using standard criteria published in the literature [4,15]: (a) foci of hyperintensity on T1W sequences and hypointensity on T2W sequences, representing haemorrhagic/proteinaceous active glandular components; (b) linear or spiculated retracting plaques with low signal intensity on all sequences, representing regions of fibrosis and lesions with smooth muscle hypertrophy. Furthermore, punctate or curvilinear signal voids on T2*WI (considered suggestive for endometriotic foci) were assessed by *Readers 1* and *3*. The results for *Reader 1* were considered the gold standard, being the most experienced reader.

### 2.4. Statistical Analysis

The data were analysed with the open-source statistical software Jamovi v. 2.2.5 (The jamovi project (2021) [Computer Software]. Retrieved from https://www.jamovi.org (accessed on 21 March 2022), sourced from Bergamo, Italy). Qualitative variables are described as frequencies and percentages. Interobserver agreement rates between *Reader 1* and *Reader 2* on a DIE location-based level were measured through Cohen’s kappa coefficient. According to Fleiss’s equally arbitrary guidelines, a k-value of less than 0.40 was considered a poor agreement; a k-value ranging from 0.40 to 0.75 was a fair to good agreement, and greater than 0.75 was an excellent agreement. Cohen’s kappa scores were also calculated to measure interobserver agreement between *Reader 1* and *Reader 3* on locations considered positive for DIE on T2*W imaging. Fisher’s exact test was used to evaluate the prevalence of signal voids assessed by Readers 1 and 3 on these sequences. A *p*-value < 0.05 was considered significant.

## 3. Results

### 3.1. Study Population

Among the 45 consecutive women, 1 patient was excluded due to poor image quality of the T2*W sequence. Therefore, the final study population included 44 patients. Patients’ ages ranged from 20 to 49 years, and the mean age was 32.7 ± 7.8 years. Forty-one patients (93.2%) had a previous TVUS examination positive for DIE, while three patients (6.8%) only had a clinical history suggestive of endometriosis. A total of 5 patients had prior surgery: 3 Caesarean sections, 1 myomectomy, and 1 salpingectomy.

### 3.2. MRI Findings

The locations and signal characteristics of the endometriotic pelvic lesions reported by the most experienced radiologist (*Reader 1*) are summarized in Table 3. Diagnosis of endometriosis was confirmed in 42 (95.4%) out of the 44 included patients by *Reader 1*. The torus uterinus, the uterine-sacral ligaments, and the ovarian peritoneal surfaces were the sites most frequently involved with DIE. Most of the locations positive for endometriosis were observed on T2W (*n* = 279), followed by T1W (*n* = 57) and T2*W sequences (*n* = 43). The most common localizations of T2*W signal voids were ovaries (*n* = 19, 44.18%) and the torus uterinus (*n* = 6, 13.95%). Examples of signal voids consistent with DIE are shown in Figure 1.

### 3.3. Agreement between Readers with Different Experience

A comparison between the MRI findings detected by *Reader 1*, who analysed the conventional MRI protocol with the addition of T2*W sequences, and *Reader 2,* who analysed only the conventional protocol, was performed, as reported in Table 4. The overall agreement between the two readers was excellent (95.9%; kappa = 0.891). The inter-observer reliability was almost perfect when only considering the most frequently involved sites: the torus uterinus (97.7%; kappa = 0.933), bilateral utero-sacral ligaments (93.3%; kappa = 0.818), ovaries (97.7%; kappa = 0.953) and ovarian peritoneal surfaces (95.5%; kappa = 0.899).

Table 5 summarizes the number and location of T2*W signal voids assessed by the most experienced reader (*Reader 1*) and the reader with less experience (*Reader 3*). The number of T2*W lesions observed by *Reader 3* (*n* = 77) was significantly higher compared with *Reader 1* (*n* = 43) (*p*-value < 0.001). The inter-observer agreement was poor (54.5%; kappa = 0.360). The sites most frequently misdiagnosed as DIE by *Reader 3* were the vaginal fornix and rectum due to intraluminal air artefacts and post-operative scars due to susceptibility artefacts caused by surgical sutures (i.e., Caesarean scars).

## 4. Discussion

Endometriosis is one of the most common gynaecological diseases, affecting one out of ten women of childbearing age. This condition leads to many nonspecific symptoms, such as chronic pelvic pain, menstrual abnormalities, dysuria, and infertility, that may overlap with other gynaecologic, urologic, and gastrointestinal disorders [1,2]. Consequently, many patients experience long delays between the onset of symptoms and the diagnosis of endometriosis, causing a significant worsening of quality of life, increasing physical and psychological morbidity, and use of health care resources [16,17]. Even if laparoscopy is the gold standard for diagnosis, MRI is considered the best noninvasive modality for the evaluation, staging, and preoperative assessment of pelvic endometriosis. Conventional MRI protocols include T1W sequences that show subacute blood products (i.e., methaemoglobin) as high signal intensity foci, allowing forthe detection of active haemorrhagic lesions. Conversely, T2*W sequences are sensitive to chronic blood by-products (i.e., haemosiderin) that are visualized as signal void artefacts due to their capacity to create localized inhomogeneity of the magnetic field. Endometriotic implants can result in haemosiderin-laden deposits caused by repeated bleeding [4,18].

For this reason, it has recently been proposed in the literature that T2*W imaging can provide added diagnostic value to conventional MRI protocols in endometriosis detection [13,14]. Some authors suggested that susceptibility-weighted imaging (SWI) can contribute to the diagnosis of endometriomas by revealing T2*W signal voids in the cyst wall [18,19]. Further studies confirmed these results, demonstrating that SWI can be effective in the differential diagnosis between endometriomas and haemorrhagic cysts [20]. Recent literature has also shown promising results for T2*W imaging in extra-ovarian endometriosis [13,21]. Pin et al. found that susceptibility-weighted sequences might improve the diagnostic performance of conventional MRI protocol in DIE assessment, with an increase in sensitivity (from 88.2% to 94.1%) and specificity (from 68.8% to 73.3%) [22]. In a recent publication, the addition of T2*W sequences to MRI protocols revealed even more favourable results: the sensitivity improved from 65.4% to 96.15%, while the specificity improved from 71.4% to 85.7% [14]. Nevertheless, in this study both readers were highly experienced radiologists, which may represent a limitation to the generalizability of the results. Moreover, no studies to date analysed how susceptibility artefacts may be a source of confusion and diagnostic overestimation for the interpreting radiologist.

Thus, T2*W sequences currently are not included in MRI conventional protocols suggested by international guidelines and their usefulness as an additional sequence for DIE detection needs to be further assessed [10,15].

Contrary to reports in the literature, our study revealed that T2*W imaging did not increase the diagnostic accuracy significantly when compared with conventional MRI protocols in assessing DIE. Firstly, artefacts caused by intestinal gas, haemorrhagic foci not linked to endometriosis (i.e., haemorrhagic corpus luteum), phleboliths, and other undesirable sources of magnetic field inhomogeneity are often indistinguishable from signal voids caused by blood by-products (Figure 2, Figure 3, Figure 4 and Figure 5). These artefactual foci of signal loss may lead to overestimating the number and size of endometriotic implants. Moreover, the majority of haemorrhagic endometriotic lesions that demonstrate T2*W signal loss also show typical high signal on the T1W fat sat images and therefore are readily detectable on conventional MRI sequences. At the same time, predominantly fibrotic implants without haemorrhagic components are easily visible on classical T2W images and do not contain areas of T2*W signal void.

As previously reported, our data confirmed that the torus uterinus, the utero-sacral ligaments, and the peritoneal surface of the ovarian fossa are the pelvic sites most frequently involved with DIE [23]. Endometriosis implants involving these structures appeared with a signal void on T2*W sequences with a relatively low frequency (17.1%, 7.5%, and 19.2%, respectively) and were easily detected on conventional sequences. For these reasons, the addition of T2* W sequences to conventional endometriosis protocols—which prolongs the exam by approximately 5 min—does not seem to be justifiable.

The present study has several limitations. First, the sample size was relatively small, and studies with larger groups are needed to support our findings. Second, the number of readers was limited, and among them, only one was highly experienced in gynaecological imaging. Third, at the time of writing this manuscript, only a few patients had undergone laparoscopic surgery after the MRI examinations. Therefore, these findings are not discussed in the study, and a correlation with the standard of reference in further analysis will strengthen the imaging results.

## 5. Conclusions

Due to its capacity to detect chronic blood degradation products, T2*-weighted imaging has many relevant clinical applications, particularly in the neuroimaging field. Moreover, this MRI technique has recently been also applied to pelvic and gynaecologic imaging. Although T2*W sequences can effectively detect haemosiderin deposits in endometriotic foci, they do not appear to add a significant contribution to either the detection or the staging of endometriosis. Furthermore, artefacts caused by additional sources of magnetic signal voids on T2*W sequences may lead to diagnostic overestimation, especially for readers with less experience.

## Figures and Tables

**Figure 1 diagnostics-12-01545-f001:**
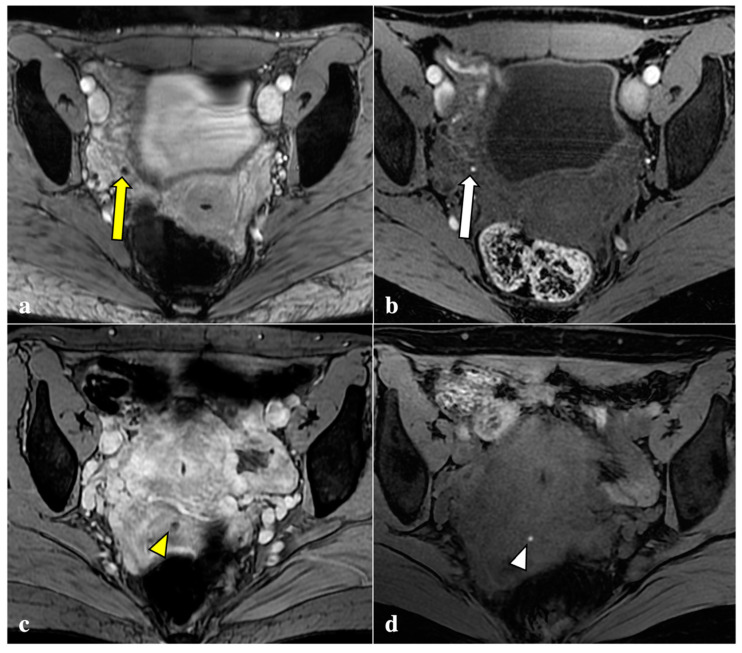
Signal voids consistent with deep endometriosis foci. (**a**,**b**): 27-year-old patient with endometriosis involving the right fallopian tube. T2*W sequence shows a signal void (yellow arrow) seen as a corresponding tiny hyperintense endometriotic haemorrhagic focus on T1W fat sat image (white arrow); (**c**,**d**): 38-year-old patient with anterior vaginal fornix endometriosis visible as a T2*W dark lesion (yellow arrowhead) and as a bright spot on the T1W image (white arrowhead), with the same clinical significance.

**Figure 2 diagnostics-12-01545-f002:**
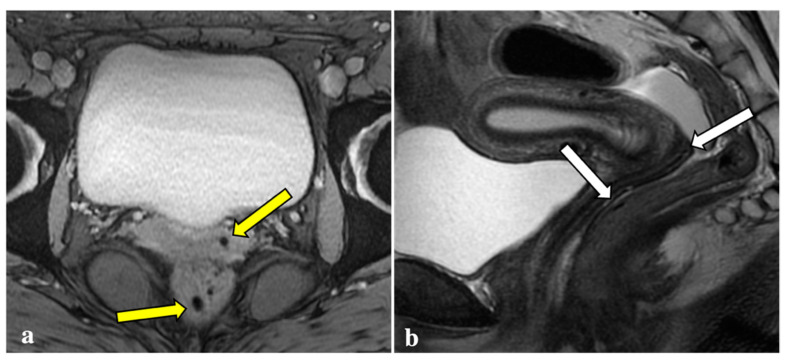
T2*-weighted imaging pitfall. Signal voids (yellow arrows) on T2*-weighted imaging (**a**) of a 24-year-old patient were subsequently related to artefacts caused by air within the vaginal fornix and the rectal lumen (white arrows), as shown in a sagittal T2W image (**b**).

**Figure 3 diagnostics-12-01545-f003:**
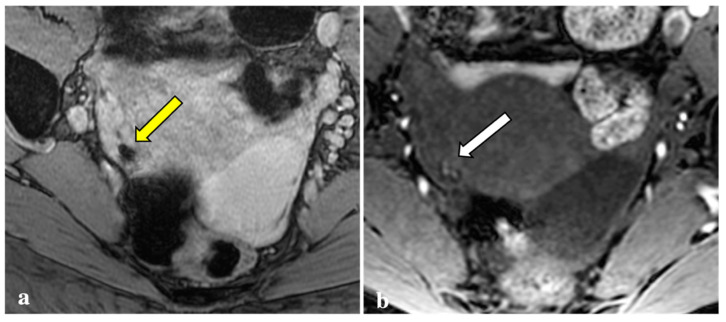
T2*-weighted imaging pitfall. An oval signal void (yellow arrow) localized in the right ovary was detected on T2*-weighted imaging in a 23-year-old patient (**a**). This lesion corresponded to a thick-walled haemorrhagic corpus luteal cyst (white arrow) with an internal high signal on T1W (**b**), consistent with blood content.

**Figure 4 diagnostics-12-01545-f004:**
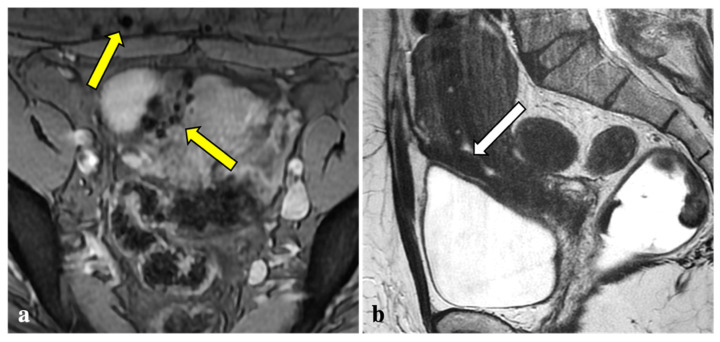
T2*-weighted imaging pitfall. Multiple T2*W signal voids along the uterine surface and abdominal wall (yellow arrows) (**a**) of a 43-year-old patient caused by surgical artefacts and caesarean scar (white arrow), as shown in a sagittal T1W image (**b**).

**Figure 5 diagnostics-12-01545-f005:**
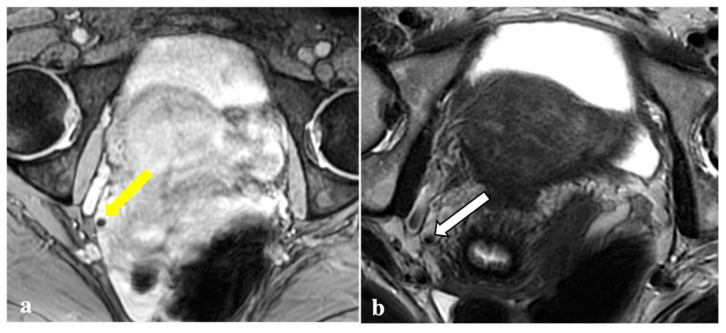
T2*-weighted imaging pitfall. A punctate signal void on T2*W (yellow arrow) (**a**) observed in a 44-year-old patient showed low signal in T2W (white arrow) (**b**) and in the other sequences (not shown). It was reported as a phlebolith by the expert reader.

**Table 1 diagnostics-12-01545-t001:** MRI protocol used in the study.

Sequence	Plane	TR/TE (ms)	FOV (mm)	Slice Thickness/Intersection Gap (mm)	Flip Angle	Nex
T2W SSFSE	axial	3100/80	320 × 320	5/1	160°	1
T2W FRFSE	axial, sagittal and coronal of the uterus	5554/102	256 × 256	3/0.3	140°	6
T1W FSPGR	axial	7.9/2.2	340 × 260	3/0	12°	4
T2*W MERGE	axial	400/5.6	320 × 320	4/0.4	20°	2

T2W: T2-weighted imaging, SSFSE: single-shot fast spin-echo, FRFSE: fast relaxation fast spin-echo, T1W: T1-weighted imaging, FSPGR: fast spoiled gradient recalled echo, T2*W: T2*-weighted imaging, MERGE: multi-echo recombined gradient echo, TR: repetition time, TE: echo time, FOV: field of view.

**Table 2 diagnostics-12-01545-t002:** Checklist of anatomical pelvic structures used by *Readers* to assess endometriosis sites.

Endometriosis Sites
*Anterior compartment*
Prevescical space
Vescicouterine/vescicocervical space
Vescicovaginal space
Round ligaments
Bladder
Ureters
*Middle compartment*
Ovaries
Ovarian peritoneal surface
Uterine serosal
Broad ligaments
Parametrium/paracolpum
Tubes
Vaginal fornix
*Posterior compartment*
Torus uterinus and retrocervical space
Utero-sacral ligaments
Rectovaginal space
Rectouterine pouch
Rectum/rectosigmoid
*Other sites*
Small bowel
Surgical scars
Abdominal/pelvic wall

**Table 3 diagnostics-12-01545-t003:** Localizations and MRI signal characteristics of endometriotic lesions detected by the most experienced observer (*Reader 1*).

Endometriosis Sites	Hypointense Lesions on T2W	Hyperintense Foci on T1W	Signal Voids on T2*W
* **Overall (n, %)** *	**279** (100)	**57** (100)	**43** (100)
*Anterior compartment*			
Prevescical space	3 (1.07)	0	0
Vescicouterine/ vescicocervical space	13 (4.66)	2 (3.51)	1 (2.33)
Vescicovaginal space	3 (1.07)	0	0
Round ligaments	13 (4.66)	0	0
Bladder	1 (0.36)	0	0
Ureters	7 (2.51)	0	0
Urachal remnants	1 (0.36)	0	0
*Middle compartment*			
Ovaries	21 (7.53)	23 (40.36)	19 (44.18)
Ovarian peritoneal surface	26 (9.32)	8 (14.03)	5 (11.63)
Uterine serosal	6 (2.15)	0	0
Broad ligaments	13 (4.66)	1 (1.75)	0
Parametrium/paracolpum	3 (1.07)	0	0
Tubes	17 (6.09)	4 (7.02)	3 (6.97)
Vaginal fornix	11 (3.94)	2 (3.51)	2 (4.65)
*Posterior compartment*			
Torus uterinus and retrocervical space	35 (12.55)	6 (10.53)	6 (13.95)
Utero-sacral ligaments	67 (24.01)	8 (14.04)	5 (11.63)
Rectovaginal space	9 (3.23)	1 (1.75)	0
Rectouterine pouch	18 (6.45)	1 (1.75)	0
Rectum/rectosigmoid	7 (2.51)	0	1 (2.33)
*Other sites*			
Small bowel	1 (0.36)	0	0
Surgical scars	2 (0.72)	1 (1.75)	1 (2.33)
Abdominal/pelvic wall	2 (0.72)	0	0

T2W: T2-weighted imaging, T1W: T1-weighted imaging, T2*W: T2*-weighted imaging.

**Table 4 diagnostics-12-01545-t004:** Comparison between MRI findings detected on the conventional protocol with the addition of T2*W imaging (*Reader 1*) and only on the conventional protocol (*Reader 2*).

Endometriosis Sites	*Reader 1* (Conventional Protocol + T2*W)	*Reader 2* (Conventional Protocol)	Agreement (%)	Kappa
* **Overall (n, %)** *	**301** (100.00)	**295** (100.00)	**95.9**	**0.891**
*Anterior compartment*				
Prevescical space	3 (1.00)	3 (1.02)	100	1.000
Vescicouterine/vescicocervical space	13 (4.32)	15 (5.08)	95.5	0.895
Vescicovaginal space	3 (1.00)	3 (1.02)	100	1.000
Round ligaments	13 (4.32)	12 (4.07)	98.9	0.953
Bladder	1 (0.33)	1 (0.34)	100	1.000
Ureters	7 (2.32)	6 (2.03)	97.7	0.910
Urachal remnants	1 (0.33)	1 (0.34)	100	1.000
*Middle compartment*				
Ovaries	38 (12.63)	36 (12.2)	97.7	0.953
Ovarian peritoneal surface	32 (10.63)	28 (9.49)	95.5	0.899
Uterine serosal	6 (1.99)	7 (2.37)	93.2	0.730
Broad ligaments	13 (4.32)	15 (5.08)	95.5	0.830
Parametrium/paracolpum	3 (1.00)	3 (1.02)	100	1.000
Tubes	17 (5.65)	14 (4.75)	96.6	0.888
Vaginal fornix	11 (3.65)	13 (4.41)	95.5	0.885
*Posterior compartment*				
Torus uterinus and retrocervical space	35 (11.63)	34 (11.53)	97.7	0.933
Utero-sacral ligaments	67 (22.26)	65 (22.03)	93.2	0.818
Rectovaginal space	8 (2.66)	9 (3.05)	97.7	0.927
Rectouterine pouch	18 (5.98)	16 (5.42)	95.5	0.904
Rectum/rectosigmoid	7 (2.32)	10 (3.39)	93.3	0.783
*Other sites*				
Small bowel	1 (0.33)	0	100	1.000
Surgical scars	2 (0.66)	1 (2.32)	100	1.000
Abdominal/pelvic wall	2 (0.66)	0	97.7	0.656

T2*W: T2*-weighted imaging.

**Table 5 diagnostics-12-01545-t005:** Comparison between regions described as positive for signal voids on T2*-weighted sequences by Readers 1 and 3.

Endometriosis Sites	Signal Voids Detected by *Reader 1*	Signal Voids Detected by *Reader 3*	Kappa	*p*-Value *
** *Overall (n, %)* **	**43** (100)	**77** (100)	**0.360**	**<0.0001**
*Anterior compartment*				
Prevescical space	0	0	-	-
Vescicouterine/ vescicocervical space	1 (2.33)	1 (1.30)	-	-
Vescicovaginal space	0	0	-	-
Round ligaments	0	0	-	-
Bladder	0	0	-	-
Ureters	0	0	-	-
Urachal remnants	0	0	-	-
*Middle compartment*				
Ovaries	19 (44.18)	25 (32.47)	0.761	<0.0001
Ovarian peritoneal surface	5 (11.63)	6 (7.79)	0.896	<0.0001
Uterine serosal	0	2 (2.60)	-	-
Broad ligaments	0	2 (2.60)	-	-
Parametrium/paracolpum	0	0	-	-
Tubes	3 (6.97)	4 (5.19)	0.788	0.003
Vaginal fornix	2 (4.65)	12 (15.59)	0.225	0.018
*Posterior compartment*				
Torus uterinus and retrocervical space	6 (13.95)	7 (9.09)	0.910	<0.0001
Utero-sacral ligaments	5 (11.63)	7 (9.09)	0.614	<0.0001
Rectovaginal space	0	0	-	-
Rectouterine pouch	0	0	-	-
Rectum/rectosigmoid	1 (2.33)	7 (9.09)	0.219	0.020
*Other sites*				
Small bowel	0	0	-	-
Surgical scars	1 (2.33)	4 (5.19)	0.377	0.001
Abdominal/pelvic wall	0	0	-	-

* *p*-value was calculated with Fisher’s exact text; - Cohen’s k or Fisher’s exact test cannot be computed because one variable was constant.

## Data Availability

The data presented in this study are available on request from the corresponding author.

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
