# Peer review of "T2*-Weighted Imaging Performance in the Detection of Deep Endometriosis among Readers with Different Experience: Comparison with Conventional MRI Sequences"

_diagnostics, 2022, doi:10.3390/diagnostics12071545_

Round 1

Reviewer 1 Report

This paper exploit the potential value of T2W imaging as complimentary method for diagnosis of endometriosis. The conclusion is T2W imaging doesn't provide significant value. However, could authors summarize in the conclusion section the particular diseases that could benefit from T2W imaging?

Could the latest development in finger printing provide diagnostic value from T2W imaging without elongating acquisition time? 

Author Response

Point 1: This paper exploit the potential value of T2W imaging as complimentary method for diagnosis of endometriosis. The conclusion is T2W imaging doesn't provide significant value. However, could authors summarize in the conclusion section the particular diseases that could benefit from T2W imaging?

Response 1: Thank you for your comment. We fully agree with this issue and decided to address it by adding more details about the application of T2*-weighted imaging in other fields in the Introduction (lines 65-68) and then recall the most relevant data in the Conclusion section (lines 357-360). We acknowledge for missing this important topic, but we decided to create a manuscript as much as possible focused on the usefulness of MRI in the detection of endometriosis foci.

Point 2: Could the latest development in finger printing provide diagnostic value from T2W imaging without elongating acquisition time? 

Response 2: Thanks for your comment. Due to its ability to simultaneously assess multiple tissue properties with one acquisition, MR fingerprinting surely represents a promising technique also in pelvic imaging exams. Furthermore, this technology has recently been proposed to reduce artifacts in T2*-weighted imaging. However, to the best of our knowledge, in literature there are still no reports about its usefulness in endometriosis MRI examinations. Unfortunately, we do not have data about it, so we did not discuss this relevant topic in the present paper. We hope to have the opportunity to do it in further research.

Reviewer 2 Report

We congratulate the authors to discuss the use of MRI in endometriosis diagnosis which is still a controversial issue.
We suggest the following:

Line 48: According to the new ESHRE guidelines on endometriosis, laparoscopy seems not be the gold standard, however in many other guidelines it is still. Authors are advised to rephrase the sentence to include this fact.

Line 55: change " not-hemorrhagic" to " non- hemorrhagic"

Line 78: the patients were suspicious of endometriosis. Did they have a histological confirmation by biopsy or surgery in their charts? if yes. where the imaging and pathology findings congruent? Can the authors give an idea about the sensitivity , specificity and PPV from their study?

Line 311: change "endometriotic implants" to "endometriosis. Implants are not staged by the disease itself has a staging. 

Author Response

Point 1: Line 48: According to the new ESHRE guidelines on endometriosis, laparoscopy seems not be the gold standard, however in many other guidelines it is still. Authors are advised to rephrase the sentence to include this fact

Response 1: Thank you for your comment. We added in the introduction section a short subparagraph about this topic (lines 48-52) and also added the 2022 ESHRE endometriosis guidelines as a reference. We apologize for the omission of these very relevant updates.

Point 2: Line 55: change " not-hemorrhagic" to " non- hemorrhagic"

Response 2: We corrected the typo; thank you (line 59).

Point 3: Line 78: the patients were suspicious of endometriosis. Did they have a histological confirmation by biopsy or surgery in their charts? if yes. where the imaging and pathology findings congruent? Can the authors give an idea about the sensitivity , specificity and PPV from their study?

Response 3: Thanks for your comment. In accordance with the aforementioned clinical practice guidelines, in our Centre, clinicians only perform diagnostic laparoscopy in limited cases. In fact, at the time of writing the manuscript, only four patients (9%) have undergone surgery/biopsy after the MRI examinations. For this reason, we decided to not discuss these findings in the present paper, and we reported this limitation in the specific subparagraph (lines 312-315). However, in all four cases laparoscopy and histological findings confirmed the presence of endometriotic lesions. Finally, due to the small sample size of patients who underwent laparoscopic surgery/biopsy considered the reference standard, we could not compute diagnostic values of T2*-weighted sequences in the detection of endometriosis foci.

Point 4: Line 311: change "endometriotic implants" to "endometriosis. Implants are not staged by the disease itself has a staging.

Response 4: Thank you, we corrected it (line 362).

Reviewer 3 Report

Good job.

Author Response

Point 1: Good job.

Response 1: Thank you for your kind comment.

Round 2

Reviewer 2 Report

Dear Authors, thank you for editing your manuscript with the proposed suggestions.

No further suggestions are done at this moment.

Regards.